# Physicochemical, Rheological, and Microstructural Properties of Low-Fat Mayonnaise Manufactured with Hydrocolloids from *Dioscorea rotundata* as a Fat Substitute

Leonardo Rojas-Martin [ID], Somaris E. Quintana [ID] and Luis A. García-Zapateiro *[ID]

Research Group on Complex Fluid Engineering and Food Rheology, University of Cartagena, Cartagena 13003, Colombia
* Correspondence: lgarciaz@unicartagena.edu.co

**Abstract:** (1) Background: In this study, the potential use of *Dioscorea rotundata* hydrocolloids was evaluated to develop low-fat mayonnaise. (2) Methods: The effect of different concentrations of hydrocolloids on the physicochemical, microstructural, and rheological properties of mayonnaise was evaluated. (3) Results: Physicochemical analyses showed pH values that were stable over time but decreased with increasing hydrocolloid concentration. The color parameters showed a decrease in luminosity and an increase in the values of a* and b* over time, which can be translated into an increase in yellow and a decrease in white, with a greater accentuation in the control sample. The rheological study allowed us to obtain a non-Newtonian flow behavior of the shear-thinning type for all samples, and the flow curves were well-fitted by the Sisko model ($R^2 \geq 0.99$). The samples had an elastic rather than viscous behavior, typical of dressings and emulsions. This indicates that the storage modulus was greater than the loss modulus ($G' > G''$) in the evaluated frequency range. (4) Conclusions: hydrocolloids from *Dioscorea rotundata* have potential as a fat substitute in emulsion-type products.

**Keywords:** mayonnaise; rheological measurements; viscoelastic properties; *Dioscorea rotundata* hydrocolloids; fat substitute





## 1. Introduction

Mayonnaises are semisolid oil-in-water emulsions generally composed of approximately 70–80% oil, salt, vinegar, egg yolk, thickeners, and flavorings; however, they are considered unhealthy foods and should not be consumed in excess due to their high-fat content [1] that may cause a high cholesterol content [2] and could increase the risk of consumers suffering from cardiovascular and cerebrovascular diseases [3,4]. Different alternatives are needed for the development of low-fat mayonnaise, i.e., the use of thickening agents such as modified starches, hydrocolloids or gums [5–7]. In this sense, the substitution of a major ingredient for a microstructure product is a challenge in the food industry, as the flavor, texture, appearance, and color attributes of the traditional product must be conferred, and in turn, prolong its useful life [1].

Several studies have shown that the use of hydrocolloids as fat substitutes reduces the negative impact on health and in turn improves the functional properties of the product, providing a creamy texture [8,9]. Novelties have also been presented in which species of hydrocolloid mixtures, such as pitch gum, Angum gum, and octenyl succinate starch, are used together with commonly reused hydrocolloids such as xanthan gum, arabic gum, and guar gum [10–13]. For mayonnaises, Santos-Fernandes and Salas-Medallo [14] replaced 15% soybean oil with chia mucilage (1%), Fiallo [15] carried out oil substitution using almond milk and iota carrageenan, Bonilla [16] reduced the oil content of conventional mayonnaise by 45% using inulin, and Evanuarini, and Susilo [17] used banana peel flour (1%) as a fat substitute to modify the rheological properties and stabilize and ensure acceptable

sensorial properties of mayonnaise. However, the use of the latter ingredients may have some sensory disadvantages, as the texture or taste of the product could be negatively affected [18,19]. Therefore, the objective is to identify alternatives of natural ingredients with technological functionality for the substitution of oils in mayonnaise.

Hydrocolloids have a high molecular weight. They have technological properties as stabilizers, thickeners, gelling agents, emulsifiers, fat substitutes, texture modifiers, clarifying agents, flocculants, mixers, edible coatings, and biofilm formers [20]. These molecules have simple and multiple branches that can interact with water molecules or other substances through hydrogen bonds or van der Waals forces [21]. Natural hydrocolloids are widely used in the food industry due to their virtues such as non-toxicity, biocompatibility, environmentally friendliness, and economics, in contrast to synthetic and semi-synthetic ingredients that do not offer the same benefits [13]. Their use as a fat substitute plays an important role because it simulates the sensorial and technological properties of fat, that is, maintaining the texture of the food and the sensation in the mouth, as well as ensuring moisture retention [22]. The rheological behavior of solution-added hydrocolloids is affected by the structural characteristics of the hydrocolloid backbone and its side chains, molecular weight, and conformation of the hydrocolloid molecules, as well as the solvent conditions [23].

*Dioscorea rotundata* is a native tuber from Africa and tropical regions [24], which stands out due to its high economic value and high nutritional value for urban and rural populations, providing approximately 200 calories in the daily diet of more than 300 million people in the tropics [25]. It also represents a good source of carbohydrates, vitamins (A, C, B1), mineral salts (calcium, iron, phosphorus), and essential amino acids (arginine, leucine, valine, and isoleucine) and has low levels of fat [26] and is therefore a good alternative ingredient with different technological properties. However, the use of hydrocolloids as technological ingredients has not been done to improve the nutritional value of food products. Therefore, the objective of this research was to investigate the feasibility of *Dioscorea rotundata* hydrocolloids as a fat substitute in mayonnaise. The physicochemical, rheological, and microstructural properties of mayonnaises with different concentrations of hydrocolloids were studied.

## 2. Materials and Methods

### 2.1. Materials

*Dioscorea rotundata* tubers, purchased from the local market in Cartagena, Colombia, were selected according to their size and shape. Xanthan gum was purchased from TEC-NAS (Medellin, Colombia). Commercial vegetable oil, spices, eggs, salt, and sugar were purchased from a local market. Olive oil, spices, eggs, salt and sugar were purchased from a local commercial distributor (Cartagena, Colombia).

### 2.2. Methods

#### 2.2.1. Obtention of Hydrocolloids from *Dioscorea rotundata*

Hydrocolloid extraction from *Dioscorea rotundata* was carried out using the methodology proposed by Quintana et al. [27], with some modifications. Initially, a solid–liquid extraction was carried out with pulp in a 1:10 ratio in acidified water at pH 3 for 4 h under magnetic stirring at 80 °C. The mixture was filtered, and the obtained solution was mixed with ethanol in a 1:1 ratio for 2 h at 4 °C to precipitate the hydrocolloids. Subsequently, centrifugation was performed and the precipitate was dried at 40 °C until a constant weight.

#### 2.2.2. Development of Mayonnaise

Different mayonnaises were developed to evaluate the *Dioscorea rotundata* hydrocolloids as a fat substitute; for this, two hydrocolloid solutions were prepared, at 15% by weight, i.e., (HDR15), and 20% wt. (HDR20), to replace the oil content at percentages of 25, 50, and 75%, obtaining seven formulations (Table 1).

**Table 1.** Low-fat mayonnaise formulation using *Dioscorea rotundata* hydrocolloids as a fat substitute.

| Sample Code | Hydrocolloid | | Oil % | Eggs % | Xanthan Gum % | Sugar % | Salt % | Vinegar % | Sodium Benzoate % | Species % | Water % |
|---|---|---|---|---|---|---|---|---|---|---|---|
| | Solution | % | | | | | | | | | |
| SC | 0 | 0 | 64 | 10 | 0.2 | 1 | 1.5 | 5.4 | 0.5 | 1 | 16.4 |
| SC-HDR15-25 | HDR15 | 16 | 48 | 10 | 0.2 | 1 | 1.5 | 5.4 | 0.5 | 1 | 16.4 |
| SC-HDR15-50 | HDR15 | 32 | 32 | 10 | 0.2 | 1 | 1.5 | 5.4 | 0.5 | 1 | 16.4 |
| SC-HDR15-75 | HDR15 | 48 | 16 | 10 | 0.2 | 1 | 1.5 | 5.4 | 0.5 | 1 | 16.4 |
| SC-HDR20-25 | HDR20 | 16 | 48 | 10 | 0.2 | 1 | 1.5 | 5.4 | 0.5 | 1 | 16.4 |
| SC-HDR20-50 | HDR20 | 32 | 32 | 10 | 0.2 | 1 | 1.5 | 5.4 | 0.5 | 1 | 16.4 |
| SC-HDR20-75 | HDR20 | 48 | 16 | 10 | 0.2 | 1 | 1.5 | 5.4 | 0.5 | 1 | 16.4 |

HDR15: *Dioscorea rotundata* hydrocolloid solution at 15%. HDR20: *Dioscorea rotundata* hydrocolloid solution at 20%.

Initially, a control sample (SC) was performed by dispersion of xanthan gum (0.2% wt.) in distilled water (16.4% wt.) under constant stirring for 20 min at room temperature to achieve a homogeneous solution. After that, eggs (10% wt.), salt (1.5%), sugar (1%), vinegar (5.4%), and sodium benzoate (0.5%) were mixed and stirred for 15 min. Subsequently, olive oil (64% wt.) was added in a continuous phase by homogenization at 7000 rpm for 5 min using an Ultra Turrax homogenizer (IKA digital T20 Ultra Turrax, Germany). Then, in order to develop mayonnaises with hydrocolloids as a fat substitute, a first step of *Dioscorea rotundata* hydrocolloid solubilization was done under constant stirring with 10 and 20% wt. in water at 60 °C for 60 min, obtaining HDR15 and HDR20; subsequently, the aqueous phase was obtained by mixing ingredients as a control sample with the solutions. Subsequently, the oil (48, 32, and 16%) was thoroughly dispersed to obtain the emulsion.

### 2.2.3. Storage Stability Analysis

The stability of the emulsions was evaluated by transferring 25 mL of the freshly prepared mayonnaise to cylindrical tubes followed by storage at 4 and 25 °C for 28 days. Then, the physicochemical and microstructural properties of the samples were evaluated during the storage period. The stability was evaluated on the basis of emulsification efficiency (E%), which is the ratio between the volume of the emulsified dispersed phase and the dispersed phase. Two different samples of each mayonnaise were measured, and each sample was measured in triplicate.

### 2.2.4. Determination of the pH of the Product

pH was determined according to the AOAC method (942.05/90.1995) [28] by direct reading in a Metter Toledo AG SG2 model digital pH meter previously calibrated using a buffer solution with a pH value of 4.0 and 7.0.

### 2.2.5. Color Measurement

The color was measured by a colorimeter (UltraScan PRO, HunterLab, America) using the L* (lightness), a* (red–green coordinates), and b* (yellow–blue coordinate) systems. The color changes (ΔE) and the whiteness index (WI) were calculated with Equations (1) and (2):

$$\Delta E = \sqrt{\left(L_m^* - L_c^*\right)^2 + \left(a_m^* - a_c^*\right)^2 + \left(b_m^* - b_c^*\right)^2} \tag{1}$$

$$WI = 100 - \sqrt{(100 - L)^2 + a^{*2} + b^{*2}} \tag{2}$$

where the subscript m represents the sample of mayonnaises formulated with solutions of *Dioscorea rotundata* hydrocolloids (HDR15 and HDR20), and c represents the control sample (SC).

### 2.2.6. Microstructural Properties

The microstructural analysis of the mayonnaises was carried out using the methodology described by Quintana et al. [29] with some modifications. A primo Star optical microscope (Carl Zeiss Primo Star Microscopy GmbH, Jena, Germany) coupled with a DCMC310 digital camera and a $100\times$ magnification lens was used to observe the internal distribution and droplet size of the samples (approx. 50 μL). The size and distribution of the droplets were analyzed with Scope Photo software (version 3.1.615) from Hangzhou Huaxin Digital Technology Co., Ltd., Zhejiang, China.

### 2.2.7. Rheological Properties

The evaluation of rheological properties was carried out following the procedures described by Quintana et al. [30] with some modifications using a modular advanced rheometer, HAAKE MARS 60 (Thermo-Scientific, MA, USA), equipped with a coaxial cylinder (inner radius 12.54 mm, outer radius 11.60 mm, barrel length 37.6 mm, GAP 5.30 mm). Steady-state viscous flow tests were performed with a shear rate between 0.01 and 100 $s^{-1}$ at 25 °C. Stress sweeps from 0.01 Pa to 1000 Pa were performed at a frequency of 1 Hz to determine the dynamic linear viscoelastic range. The frequency sweep was carried out in the range of 0.01 to 100 rad·$s^{-1}$, using stress values that were within the linear viscoelastic range.

### 2.2.8. Statistical Analysis

All measurements were performed in triplicate. Subsequently, the data were processed using the statistical software STATGRAPHICS Centurion XVI Version 16.1.11. Descriptive statistics were used to determine the mean value and standard deviation; additionally, analysis of variance (ANOVA) was performed at a significance level of 95%, using the HSD Tukey test.

## 3. Results

### 3.1. Physicochemical Properties

Low-fat mayonnaise was prepared using *Dioscorea rotundata hydrocolloids* with 25, 50, and 75% fat substituted with HDR15 solution (SC-HDR15-25, SC-HDR15-50, and SC-HDR15-75) and HDR20 (SC-HDR20-25, SC-HDR20-50, and SC-HDR20-75); a control sample (SC) was prepared with 64% oil (Table 1). The mayonnaise obtained was stable and maintained its physical characteristics under storage at 4 and 25 °C, which was related to the homogenization process and the reduction in droplet size to improve the entanglement for the stability of the emulsion.

Mayonnaise is a relatively acidic product with a long shelf life for microbial and consumer acceptance [31]. The samples have low pH values between 4.64 and 5.42, related to the vinegar content, which increases the acidity and consequently decreases the pH value; the undissociated acetic acid molecules that penetrate the cell membrane reduce the microbial stability of mayonnaise [32]. Control samples had higher pH values (5.42) than low-fat mayonnaise (4.46 to 5.23). Substitution of the oil with *Dioscorea rotundata* hydrocolloids caused a slight change in pH; this phenomenon was directly related to the proportion of hydrocolloids and oil. The mayonnaise prepared with HDR15 had a lower pH value than that prepared with HDR20. The evolution of mayonnaise during the storage time revealed that the pH of the samples was quite similar between the control and the low-fat mayonnaise (Figure 1), although the study was carried out under different temperature conditions, and no significant variations in pH were observed on the measurement days ($p > 0.05$). Replacement of oil with hydrocolloids has a positive effect on the conservation of the product and improves the reduction of oxidative deterioration caused by polyunsaturated fatty acids [33].

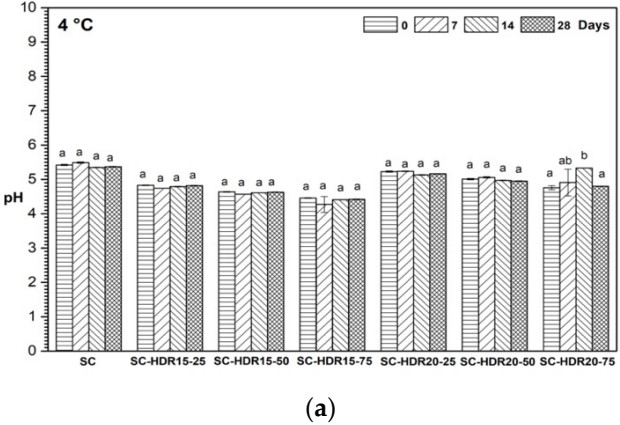
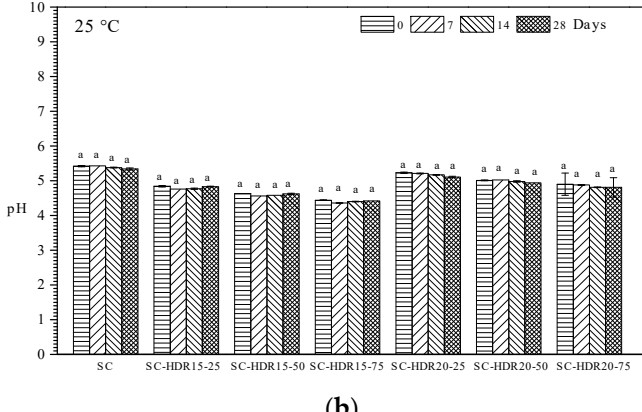

(**a**)　　　　　　　　　　　　　　　　　(**b**)

**Figure 1.** pH analysis of mayonnaise using *Dioscorea rotundata* hydrocolloids as a fat substitute (**a**) stored at 4 °C and (**b**) at 25 °C. The different letters above each bar indicate significant differences ($p < 0.05$).

### 3.2. Color Analysis

Mayonnaise color analysis was performed by evaluating the coordinated luminosity (L*), red/green (a*), and yellow/blue (b*). Moreover, the white index (WI) and change of color (ΔE) between the control (SC) and mayonnaises with HDR15 and HDR20 during storage were assessed. The concentration of hydrocolloids influenced the color parameters (Table 2): SC had L* values of 80.69 ± 2.46. Significant differences ($p < 0.05$) were obtained for mayonnaises prepared with HDR15, with values of 82.13 ± 2.95, 78.36 ± 3.89, and 74.95 ± 0.62 for SC-HDR15-25, SC-HDR15-50, and SC-HDR15-75, respectively. More similar values were obtained between the control sample and mayonnaise prepared with HDR20 (79.45 ± 5.39, 81.83 ± 5.06, and 80.44 ± 3.41 for SC-HDR20-25, SC-HDR20-50, and SC-HDR20-75, respectively). These values are attributed to the carotenoids present in the oil, responsible for the light golden color, depending on the concentration of oil in each formulation [34].

**Table 2.** Physicochemical and droplet size analyses of low-fat mayonnaise using *Dioscorea rotundata* hydrocolloids as a fat substitute.

| Control Samples | pH | L* | a* | b* | WI | ΔE | Droplet Size μm |
|---|---|---|---|---|---|---|---|
| SC | 5.42 ± 0.02 [b] | 80.69 ± 2.46 [a] | −3.18 ± 0.01 [a] | 6.74 ± 0.04 [a] | 79.29 ± 2.31 [a] | – | 1.17 ± 0.08 [a] |
| SC-HDR15-25 | 4.83 ± 0.01 [b] | 82.13 ± 2.95 [ab] | −3.70 ± 0.56 [a] | 11.53 ± 3.00 [a] | 78.23 ± 2.54 [a] | 6.52 ± 1.17 [a] | 1.18 ± 0.35 [a] |
| SC-HDR15-50 | 4.64 ± 0.01 [a] | 78.36 ± 3.89 [a] | −1.60 ± 0.01 [a] | 5.12 ± 2.01 [b] | 77.25 ± 3.74 [a] | 4.70 ± 3.68 [b] | 1.20 ± 0.17 [a] |
| SC-HDR15-75 | 4.46 ± 0.01 [a] | 74.95 ± 0.62 [b] | −3.28 ± 1.86 [a] | 5.12 ± 2.01 [b] | 74.12 ± 0.52 [b] | 3.57 ± 1.29 [b] | 1.89 ± 0.45 [c] |
| SC-HDR20-25 | 5.23 ± 0.02 [a] | 79.45 ± 5.39 [a] | −3.46 ± 0.87 [a] | 8.36 ± 1.46 [a] | 77.52 ± 5.51 [ab] | 6.08 ± 2.49 [a] | 1.31 ± 0.20 [b] |
| SC-HDR20-50 | 5.01 ± 0.01 [ab] | 81.83 ± 5.06 [a] | −2.65 ± 0.03 [a] | 5.22 ± 0.06 [b] | 80.87 ± 4.83 [a] | 2.73 ± 1.38 [b] | 1.47 ± 0.01 [b] |
| SC-HDR20-75 | 4.76 ± 0.06 [c] | 80.44 ± 3.41 [a] | −0.94 ± 0.57 [a] | 6.76 ± 1.88 [b] | 79.14 ± 2.56 [ab] | 3.68 ± 0.45 [b] | 1.65 ± 0.20 [c] |

Results are expressed as the mean ± standard deviation. The different letters within each column are significantly different ($p < 0.05$).

Regarding a*, positive values represent red and negative values indicate green [35]. It was observed that all samples presented negative results that indicated a trend towards a green color, with the most pronounced values for SC-HDR15-25%: −3.70 ± 0.56; SC had values of −3.18 ± 0.01, and similar values were obtained for samples with 75 and 25% oil substitution (SC-HDR15-75 and SC-HDR20-25: −3.28 ± 1.86 and −3.46 ± 0.87, respectively). For the parameter b*, representing a yellow color with positive and negative data for blue [35], it was found that all mayonnaises presented positive results, indicating the presence of yellow color, and this remained between 11.53 ± 3.0 for the SC-HDR15-25 sample and 5.12 ± 2.01 for the SC-HDR15-75 sample. The whiteness index (WI) of mayonnaises with 25 and 50% oil substitution did not present significant differences

($p > 0.05$) while samples with 75% substitution showed significant differences ($p < 0.05$), as shown in Table 2. The whiteness of mayonnaise is an essential quality parameter that affects consumer appeal and overall product acceptability [36,37]. The total color difference ($\Delta E$) was lower in samples with 50 and 75% oil substitution ($p < 0.05$), indicating that the sample was the lightest among all samples; therefore, the substitution with *Dioscorea rotundata* hydrocolloids did not cause unpleasant color properties for consumers.

The evolution of the color parameters over the storage period at 4 and 25 °C is presented in Figure 2. The luminosity (L*) had a progressive decrease that was greater at a temperature of 25 °C, and the control sample had the lowest values. Figure 2a,b shows the changes in L*, for each formulation, according to the day of measurement ($p < 0.05$), where SC had the highest value; at 25 °C, the samples showed a greater decrease while 4 °C preserved the value of L*. However, despite the observed reduction in luminosity, the addition of HDR15 improved stability by allowing less loss of luminosity. This is due to the presence of a branched structure of hydrophilic macromolecules that allows the formation of homogeneous emulsions with smaller oil droplets, which generates greater light scattering [38]. Rondon et al. [39] obtained similar results, showing luminosity values with the same degradation trend according to the storage time and temperature.

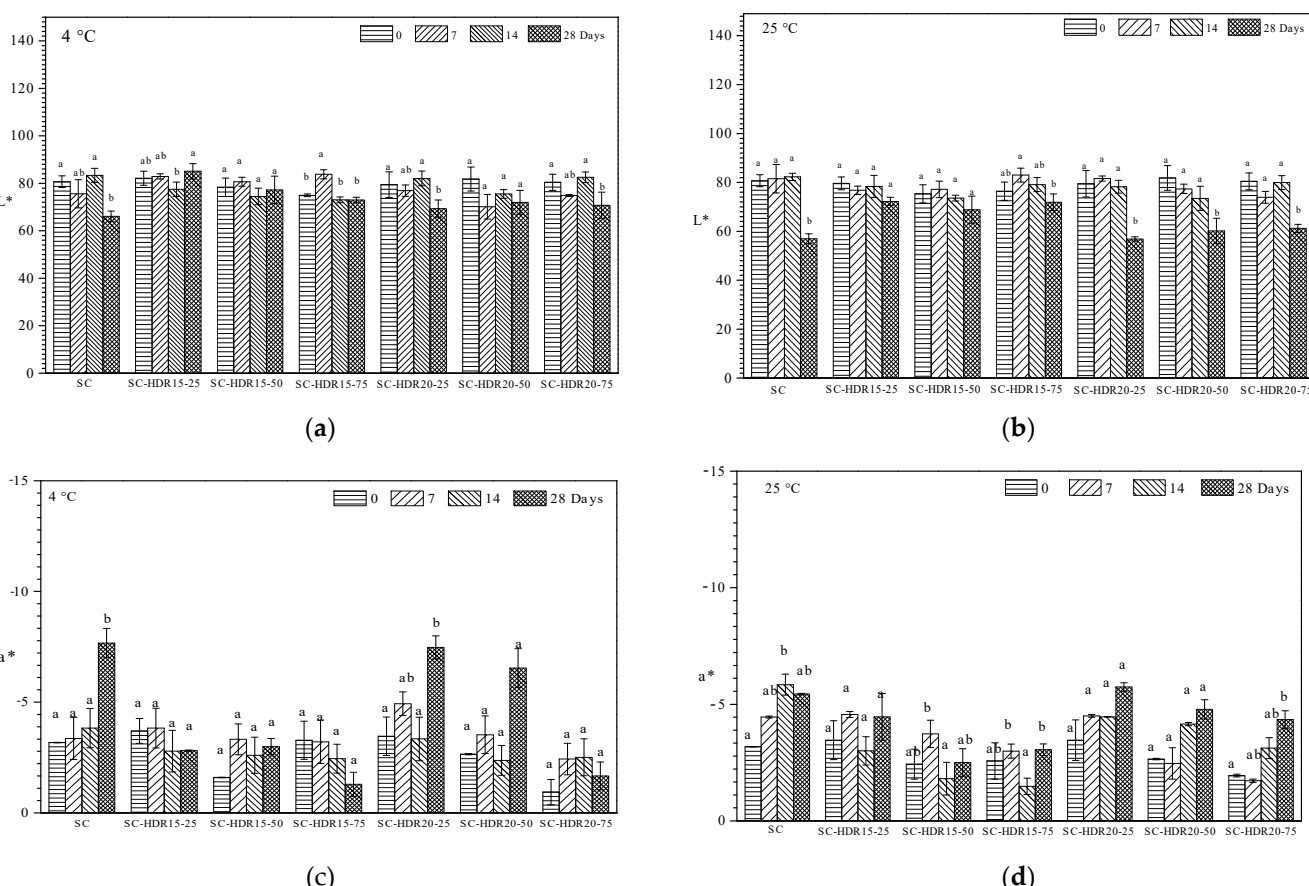

**Figure 2.** *Cont.*

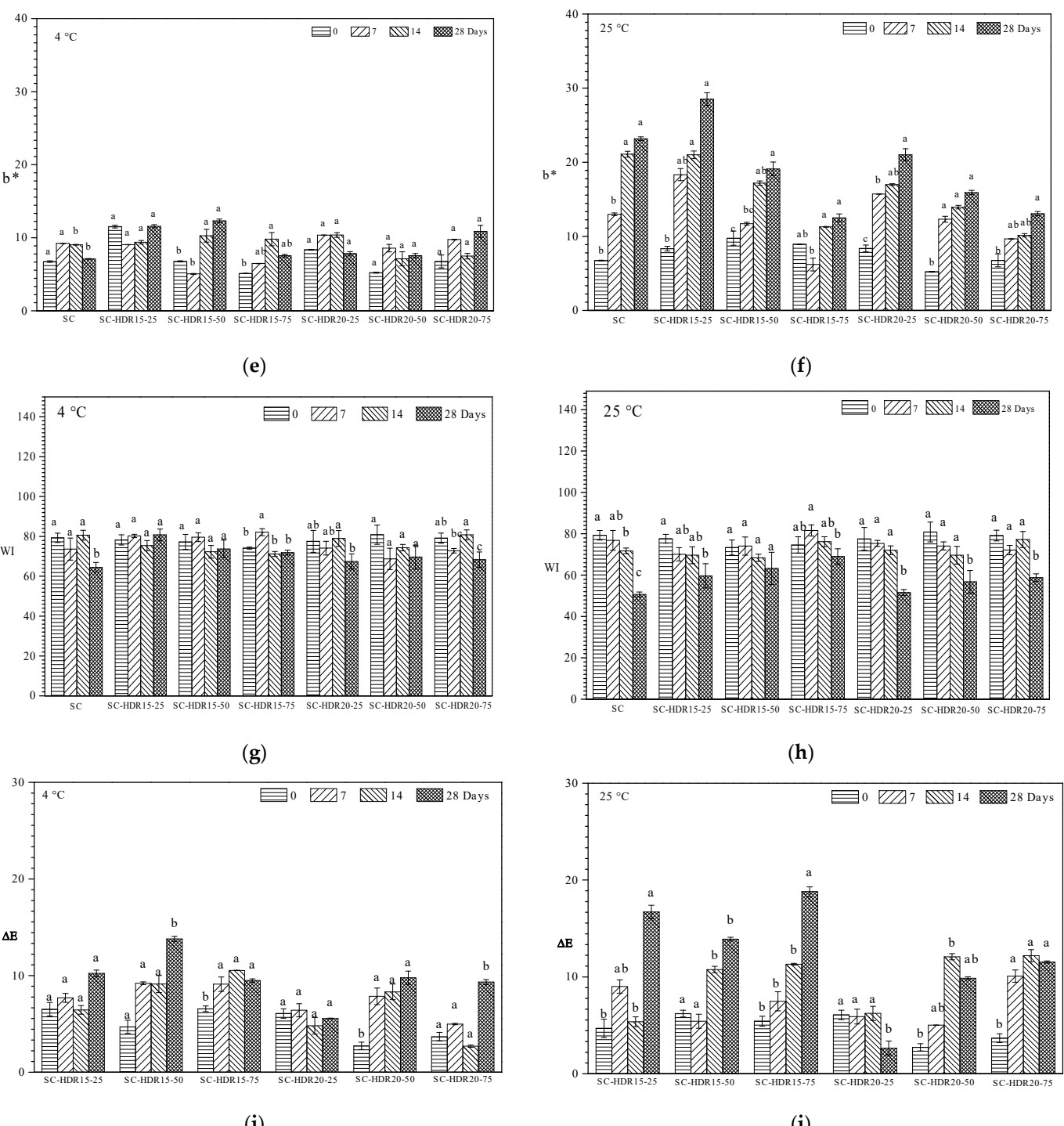

**Figure 2.** Color analysis of mayonnaise using *Dioscorea rotundata* hydrocolloids as a fat substitute: (**a**) Luminosity (L*) of samples at 4 °C, (**b**) Luminosity (L*) of the samples stored at 25 °C, (**c**) red–green coordinate (a*), storage at 4 °C, (**b**) Luminosity (L*) of samples stored at 25 °C, (**c**) red–green coordinate (a*), storage at 4 °C, (**d**) red–green coordinate (a*), storage at 25 °C, (**e**) yellow–blue coordinate (b*) of the storage of the samples at 4 °C, (**f**) yellow–blue coordinate (b*) of the samples stored at 25 °C, (**g**) whiteness index (WI) of the samples stored at 4 °C, (**h**) whiteness index (WI) of the samples stored at 25 °C, (**i**) color changes (ΔE) of the samples stored at 4 °C and (**j**) color changes (ΔE) of the samples stored at 25 °C. The different letters above the bars indicate significant differences (*p* < 0.05).

Figure 2c,d shows the behavior of parameter a* at 4 and 25 °C, respectively. An increase in negative values with a tendency to green was observed with a greater enhancement on day 28. The SC control sample had significant values of $-7.67 \pm 0.66$ on day 28 at 4 °C compared to HDR15 samples that had better stability, with values of $-2.81 \pm 0.03$, $-2.99 \pm 1.37$, and $-1.29 \pm 0.55$ for SC-HDR15-25, SC-HDR15-50, and SC-HDR15-75, respectively; for samples with HDR20, similar results were obtained to the control mayonnaise that had lower stability, with values of $-7.47 \pm 1.52$, $-6.54 \pm 3.88$, and $-1.67 \pm 1.63$, for SC-HDR20-25, SC-HDR20-50, and SC-HDR20-75, respectively.

The values of b* (Figure 2e,f) tended to increase with the storage time ($p > 0.05$) in all cases (4 and 25 °C), with a more pronounced increase in samples stored at 25 °C. According to similar studies, there were similar changes in the values of a* and b*. However, unlike in the present study, there was an increase in the lightness of mayonnaise due to the encapsulation of a soy protein isolate–pectin complex [40]. A study conducted by Yüceer et al. [41] on low-cholesterol mayonnaise presented different results, where the control mayonnaise had lower color values compared to modified samples.

The storage of the samples at 25 °C showed a greater decrease over time compared with 4 °C; a pronounced decrease was observed at 28 days; then, the samples were preserved in WI until 14 days at the temperatures evaluated. The decrease in WI (Figure 2g,h) can be understood as a decrease in quality, where the SC control sample experienced the lowest result with a value of $64.49 \pm 2.40$, showing significant differences ($p < 0.05$) compared to the samples with HDR15, with values of $80.78 \pm 2.89$, $73.70 \pm 4.45$, and $71.89 \pm 1.21$ for SC-HDR20-25, SC-HDR20-50, and SC-HDR20-75, respectively. The mayonnaises with HDR20 presented similar results to SC, with values of $67.43 \pm 3.81$, $69.62 \pm 6.05$, and $68.37 \pm 3.86$ for SC-HDR20-25, SC-HDR20-50, and SC-HDR20-75, respectively. These results demonstrate that HDR15 had a greater ability to preserve quality. The barrier or branched structure of the hydrocolloids is believed to have influenced the resistance to the oxidation initiation process or hydroperoxide formation, unlike the control sample [40]. The values of the total color difference ($\Delta E^*$) presented in Figure 2i,j increased for all samples during storage; days 14 and 28 had significantly higher values ($p < 0.05$), but the differences between the SC control and the different HDR treatments were not significant. This behavior is attributed to a decrease in stability with a tendency to phase separation [42].

### 3.3. Microstructural Properties

Figures 3 and 4 present the micrographs of low-fat mayonnaises using *Dioscorea rotundata* as a fat substitute at 4 and 25 °C, respectively. Mayonnaises had spherical drops; however, some agglomerations influenced stability.

All samples had polydisperse properties, showing different droplets, which is normal when handling food emulsions. All samples exhibited a qualitatively similar appearance, where the drops were organized in flocs, related to the homogenization process, which produces a reduction in the drop size [43]. When evaluating the evolution of the morphology with the storage time, a certain tendency of the drops towards flocculation was observed due to the association between the drops, which was attributed to a more attractive interaction due to van der Waals forces, hydrophobic forces, and exhaustion, which overcome repulsive interactions [44]. However, the micrographs showed that droplets separated into SC-HDR15-75 and SC-HDR20-75 formulations on days 16 and 28 (Figures 3 and 4). This can be attributed to an increase in repulsive interactions due to the use of mixed emulsifiers (hydrocolloids–oil) [44].

Mayonnaise has droplets that increase with time due to the fusion of one with another, causing the phenomenon of coalescence, produced by the fusion of two or more droplets to form a single droplet; this effect causes the droplets to settle due to the increased size [45]. Regarding the drop size, mayonnaise had values between $1.17 \pm 0.08$ and $3.01 \pm 0.93$ $\mu$m. SC had the largest droplet size ($3.01 \pm 0.93$ $\mu$m) followed by SC-HDR15-25 ($1.18 \pm 0.35$), SC-HDR15-50 ($1.20 \pm 0.17$), SC-HDR20-25 ($1.31 \pm 0.20$), SC-HDR20-50 ($1.47 \pm 0.10$), and SC-HDR20-75 ($1.65 \pm 0.20$), and the lowest value was obtained by SC-HDR20-75% ($1.89 \pm 0.45$).

The addition of hydrocolloids decreased the droplet size, associated with the generation of a better structure and greater entanglements due to the increased interactions between droplets [46,47]. According to Drew [48], the formation of smaller droplets prevents the creation of cremation or affects the sedimentation speed and the breaking of the emulsion.

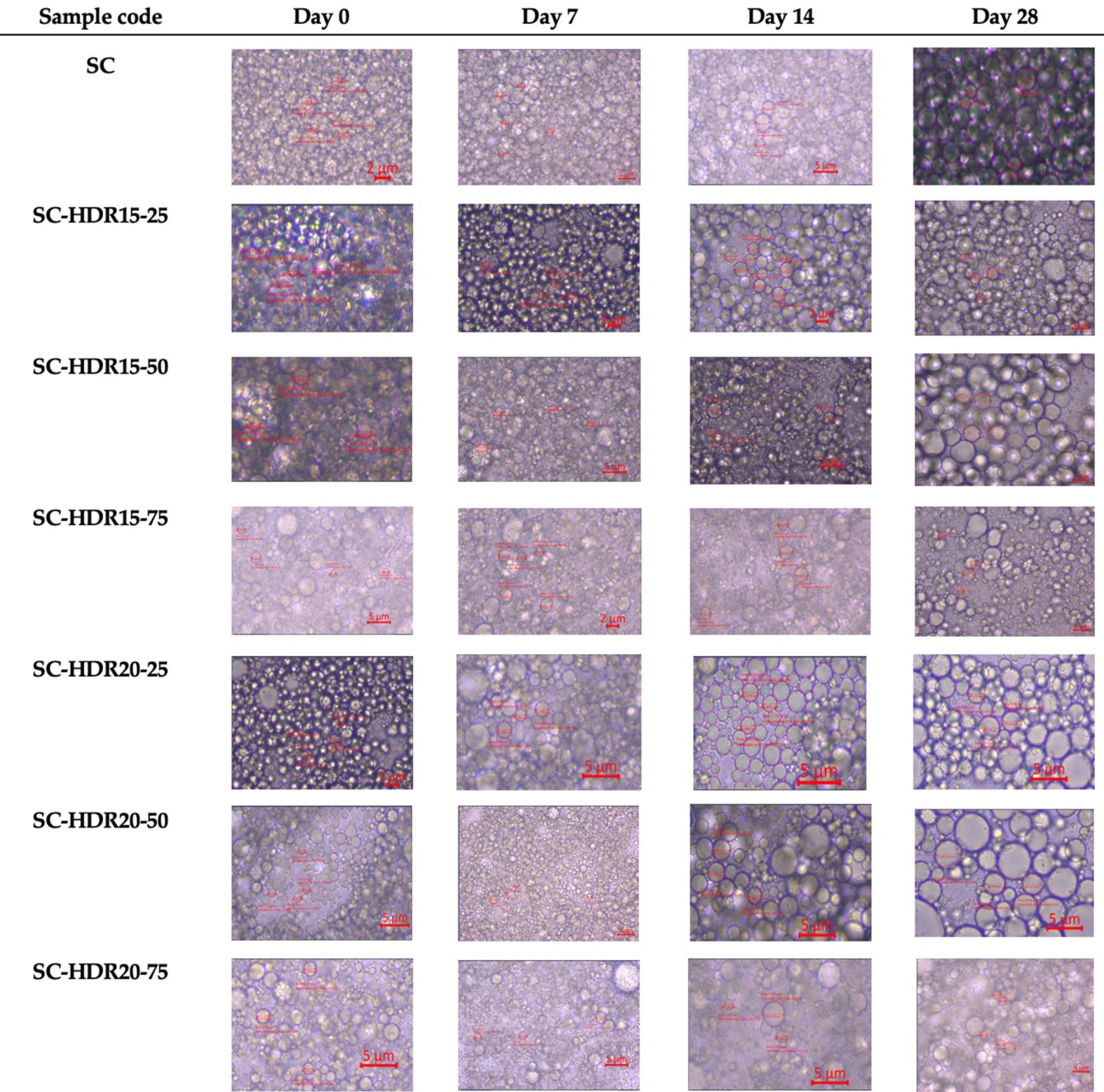

**Figure 3.** Micrographs of mayonnaise using the hydrocolloids of *Dioscorea rotundata* as a fat substitute under storage at 4 °C.

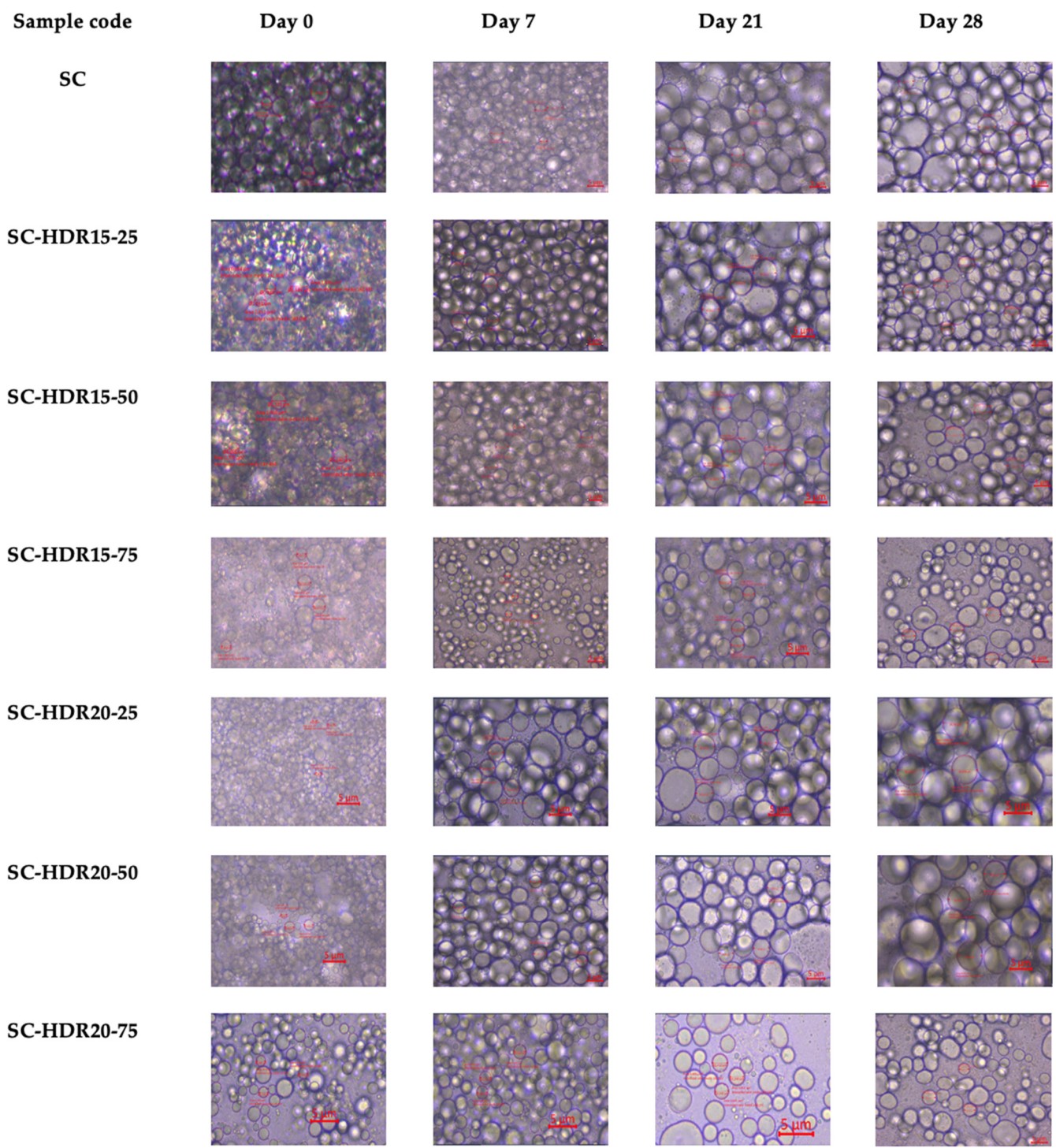

**Figure 4.** Micrographs of mayonnaise using the hydrocolloids of *Dioscorea rotundata* as a fat substitute under storage at 25 °C.

The evaluation of the emulsions based on time and a storage temperature of 4 °C and 25 °C, presented in Figure 5a,b, respectively, showed a progressive increase in the microstructure size over time. For samples at 4 °C, on day 0, the smallest size was presented, with $1.17 \pm 0.08$ μm for SC, showing small differences in the HDR15 treatments ($1.18 \pm 0.35$ μm, $1.20 \pm 0.17$ μm, and $1.89 \pm 0.45$ μm for SC-HDR15-25, SC-HDR15-50, and SC-HDR15-75, respectively) and in those with HDR20 ($1.31 \pm 0.20$ μm, $1.47 \pm 0.01$ μm, and $1.65 \pm 0.20$ μm for SC-HDR20-25, SC-HDR20-50, and SC-HDR20-75, respectively). On

day 28, the droplet size was significantly larger ($p < 0.05$) in all treatments, resulting in a decrease in stability over time. The treatments exhibited the best stability (2.47 ± 0.24 μm and 2.94 ± 0.52 μm for SC-HDR20-25 and SC-HDR20-75, respectively), unlike the control sample SC and SC-HDR15-50, which were higher, with values of 3.01 ± 0.93 μm and 5.37 ± 0.44 μm, respectively. The size of the samples stored at room temperature in Figure 5b was significantly larger ($p < 0.05$), with a range between 1.18 ± 0.35 μm and 6.15 ± 0.54 μm, with lower stability on day 28.

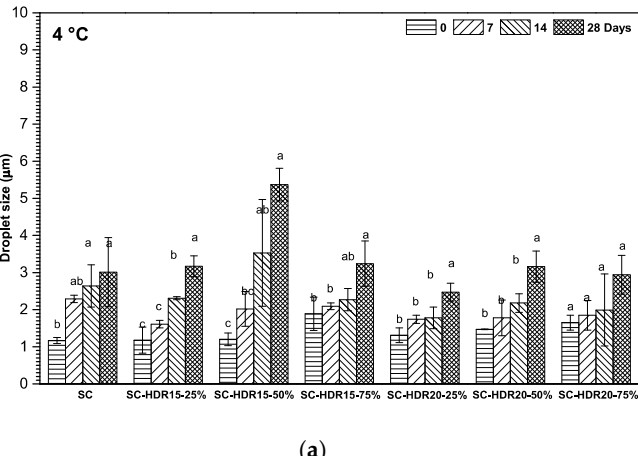

(a)

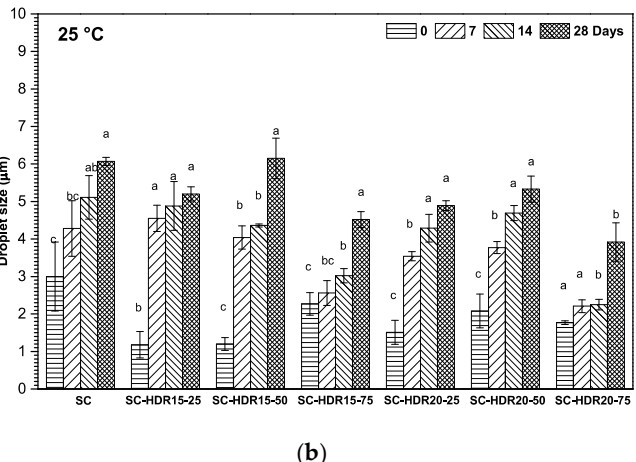

(b)

**Figure 5.** Evolution of the size of the droplets of mayonnaise using the hydrocolloid of *Dioscorea rotundata* as a fat substitute at (**a**) 4 °C and (**b**) 25 °C. The different letters above the bars indicate significant differences ($p < 0.05$).

The storage of the samples at 4 °C had better stability over time because the temperature had a direct effect on the interaction between the compounds present in the different samples, affecting the viscosity and stability of the product over time, and making them vulnerable to phenomena such as coalescence, flocculation, and droplet cremation [1]. Different investigations [49–51] on emulsions revealed that the size of the droplets in an emulsion changes with the variation in the temperature and the HLB value (hydrophilic/lipophilic balance) of the emulsifiers; the smaller the diameter of the droplets, the more likely the emulsion will be unstable after coalescence.

### 3.4. Rheological Properties

3.4.1. Stationary State

The viscous flow curves of mayonnaise are shown in Figure 6; in all cases, the viscosity decreased with increasing shear rate, indicating a flow behavior with non-Newtonian-type shear thinning [52] associated with a progressive increase in the shear rate that generated irreversible processes (coalescence, droplet bursting) and reversible processes (flocculation–deflocculation) as a response due to a gradual orientation of molecules in the direction of flow to reduce friction and the deformation of hydrated hydrocolloids [29,53,54]. The flow properties of mayonnaise depend mainly on hydrodynamic forces that dominate weak attractive forces and the movement between particles [55]. At higher shear rates, the viscosity reached constant values because all oil droplets and other molecules were mostly broken up. Therefore, only small and individual particles remained in the system [29]. Another important aspect is the positive effect on improving the consistency of the emulsion due to the addition of hydrocolloids that have hydrostatic properties to retain and stabilize moisture [56,57].

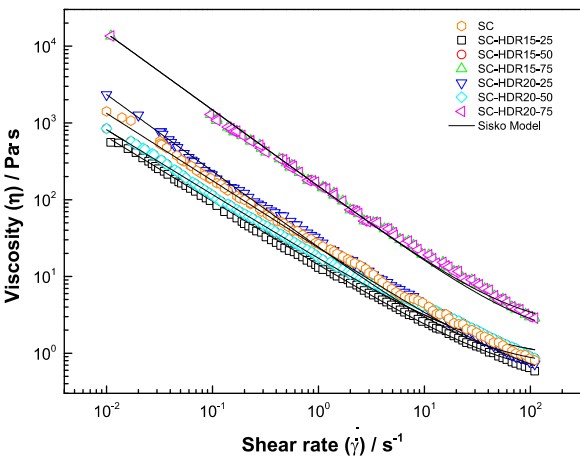

**Figure 6.** Viscous flow behavior of mayonnaise using the hydrocolloids of *Dioscorea rotundata* as a fat substitute.

For all samples, the apparent viscosity was closely related to the varying concentrations of fat-substituted hydrocolloids in the molecular structure of each suspension. An increase in viscosity was observed at higher concentrations of *Dioscorea rotundata* hydrocolloids due to the formation of a three-dimensional network in response to the interaction of the continuous medium and the oil droplet interface mixed with hydrocolloids [1]. As the shear rate increased, the mayonnaises showed characteristic behavior; for example, at a lower speed, there was a coiling or molecular entanglement in the structure, generating higher viscosity, but as the shear rate increased, a potential drop zone was generated in the curve, indicating the breakage of such bonds, leading to a decrease in viscosity until it reached a constant value [58]. The SC-HDR20-25 and SC-HDR20-50 formulations were the most similar to SC. Thus, SC-HDR15-25 mayonnaise had the lowest viscosity. This shows that the rheological parameters are under the influence of the percentage, concentration, and interaction of the applied hydrocolloid substitution with other hydrocolloids such as xanthan gum [2,59–61]. Similar behaviors have been shown in reduced-fat mayonnaise, i.e., reduced-fat mayonnaise using 4$\alpha$GTase-modified rice starch and xanthan gum [1], mayonnaise with antioxidants [62], and vegan egg-free mayonnaise [63].

Due to the viscous behavior of the samples, the Sisko model (Equation (3)) was used to fit the experimental data ($R^2 > 0.97$):

$$\eta = \eta_\infty + k\gamma^{n-1} \tag{3}$$

where the value of $\eta$ corresponds to the apparent viscosity (Pa·s), $\eta_\infty$ is the performance index, k is the coefficient of consistency of the material, and n is the flow behavior index.

The Sisko fit parameters are listed in Table 3. In all cases, n was less than 1, corroborating the shear thinning behavior, and n decreased with the percentage of oil substitution (SC, the control sample, had the lowest value). The lower values of n represent a higher slope of the curve, which generates a more pronounced decrease in viscosity as the shear rate increases due to the generation of a potential drop zone in the curve, indicating the breakage of molecular bonds, leading to a decrease in viscosity until a constant value [58,64]. $\eta_\infty$ increase with the percentage of oil substitution with hydrocolloids from *Dioscorea rotundata.* This shows that the rheological parameters are under the influence of the percentage, concentration, and interaction of the hydrocolloid substitution applied with other hydrocolloids [2,59–61]. The highest values were presented by the samples with the highest HDR substitution, that is, SC-HDR15-75 and SC-HDR20-75.

**Table 3.** Sisko adjustment parameters and viscoelastic properties at 10 rad·s$^{-1}$ of mayonnaise using the hydrocolloid *Dioscorea rotundata* as a fat substitute.

| Sample Code | k Pa·s$^n$ | n | $\eta_\infty$ Pa·s | $R^2$ |
|---|---|---|---|---|
| SC | 23.36 ± 2.48 [c] | 0.12 ± 0.02 [a] | 0.31 ± 5.23 [d] | 0.97 |
| SC-HDR15-25 | 13.80 ± 0.56 [a] | 0.16 ± 0.00 [b] | 0.40 ± 1.04 [d] | 0.99 |
| SC-HDR15-50 | 16.50 ± 1.01 [ab] | 0.15 ± 0.01 [b] | 0.80 ± 1.95 [b] | 0.99 |
| SC-HDR15-75 | 147.37 ± 0.72 [d] | 0.14 ± 0.01 [b] | 1.96 ± 0.01 [a] | 0.99 |
| SC-HDR20-25 | 23.48 ± 0.12 [c] | 0.12 ± 0.01 [a] | 0.64 ± 0.01 [cd] | 0.99 |
| SC-HDR20-50 | 16.50 ± 1.01 [ab] | 0.15 ± 0.01 [b] | 0.80 ± 1.95 [b] | 0.99 |
| SC-HDR20-75 | 148.66 ± 0.74 [d] | 0.14 ± 0.01 [b] | 1.31 ± 0.01 [a] | 0.99 |

Results are expressed as the mean ± standard deviation. The different letters within each column are significantly different ($p < 0.05$).

This behavior is attributed to the formation of a three-dimensional network in response to the interaction between the continuous medium and the interface of oil droplets mixed with hydrocolloids [1]. K decreased with the percentage of oil, whereas mayonnaise with HDR15 had lower values than HDR20. The control sample had an intermediate value, while the highest values were the SC-HDR15-75 and SC-HDR20-75 samples. Greater consistency was evidenced in the samples with a higher percentage of HDR substitution. This is because there is greater water droplet trapping due to an increase in the density and structure of the emulsions [64]. The use of *Dioscorea rotundata* hydrocolloids as a fat substitute in different concentrations stands out as an alternative to the production of microstructured foods. Samples with a low concentration of hydrocolloids showed low resistance of the fluid against the flow, resulting in low viscosity. On the contrary, SC-HDR15-75 and SC-HDR20-75 emulsions with a higher concentration of hydrocolloids had better characteristics, with a higher viscosity and a high resistance of the fluid against flow due to a better droplet distribution [29]. According to the above, it is concluded that a smaller droplet size contributes to a higher viscosity in emulsions with *Dioscorea rotundata* hydrocolloid as a fat substitute. This leads to a direct relationship between the particle size and the shear-thinning behavior of mayonnaise.

### 3.4.2. Viscoelastic Properties

Figure 7 shows the storage ($G'$) and the loss modulus ($G''$) in the function of the angular frequency. $G'$ was higher than $G''$ within the frequency range, predominating a solid behavior indicating that the mayonnaise had a gel-like structure of an entangled and flocculated network with a tendency to elastic solid behavior typical of dressings and emulsions [19]. Previous studies showed that mayonnaise samples had weak gel-like properties within the frequency range of 0.1 to 10 Hz [38,65]. The investigation of Ma and Barbosa-Canovas [66] reported that emulsions with higher fat concentrations had a higher $G'$ because this module represents the energy that is stored when the material is subjected to deformation. However, our study presented results similar to those of Mun et al. [1], where the emulsions that had a greater substitution of hydrocolloids as a partial substitute for oil and xanthan gum had higher values of $G'$. Specifically, it can be seen that the formulations with the highest concentration of hydrocolloids were SC-HDR15-25, SC-HDR15-50, and SC-HDR15-75. This effect could be attributed to the hydrocolloids, which, because they had a higher concentration, strengthened the gel structure of the emulsion.

However, all samples showed a plateau zone characteristic of systems such as weak gels, which exhibited similar viscoelastic characteristics [67]. Similar results have been reported for mayonnaise formulated with frozen-thawed egg yolk [68]. The development of a plateau zone was observed in some samples, characterized by $G'$ values that were almost constant with the frequency, a characteristic of quasi-elastic behavior. Nevertheless, some differences were observed depending on the percentage of *Dioscorea rotundata* hydrocolloids. Mayonnaises showed $G'$ higher than $G''$ throughout the frequency range,

indicating the proximity of the transition region in the mechanical spectrum, except for the low-frequency values of SC-HDR15-50 and SC-HDR20-75 associated with the total amount of hydrocolloids and the interaction with gums and egg yolk protein.

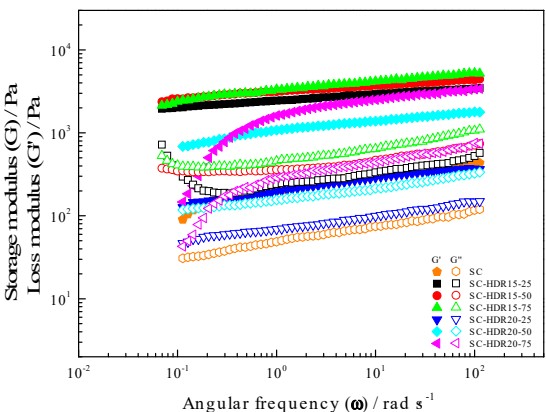

**Figure 7.** Storage and loss modulus of mayonnaise using the hydrocolloid of *Dioscorea rotundata* as a fat substitute.

This behavior can be analyzed by the power-law model (Equations (4) and (5)):

$$G' = k' \cdot \omega^{n'} \tag{4}$$

$$G'' = k'' \cdot \omega^{n''} \tag{5}$$

where $n'$ and $n''$ represent the slopes of the relationships between the modulus and frequency; $k'$ and $k''$ represent the magnitude of $G'$ and $G''$, respectively. Table 4 shows the parameters of the power-law equation that describe the modules' dependence on the oscillation frequency. In general, for SC, SC-HDR15-25, SC-HDR15-75, SC-HDR20-25, and SC-HDR20-50, the storage ($G'$) and loss ($G''$) modules were fitted to a power-law model, $R^2 > 0.96$, (Equations (4) and (5)); for SC-HDR15-50, $G''$ was adjusted, but SC-HDR20-75 was not due to the start of the plateau zone that characterized an increase in the modulus with the frequency.

**Table 4.** Viscoelastic properties of mayonnaise using the hydrocolloids of *Dioscorea rotundata* as a fat substitute.

| Sample Code | $k'$ | $n'$ | $R^2$ | $k''$ | $n''$ | $R^2$ | $\mathrm{Tan}\delta_{\omega=10\mathrm{rad}\cdot\mathrm{s}^{-1}}$ |
|---|---|---|---|---|---|---|---|
| SC | 205.35 ± 2.22 [c] | 0.17 ± 0.00 [d] | 0.98 | 48.30 ± 0.45 [a] | 0.18 ± 0.00 [b] | 0.98 | 0.24 [d] |
| SC-HDR15-25 | 2425.00 ± 10.03 [d] | 0.07 ± 0.00 [a] | 0.98 | ** | ** | ** | 0.12 [a] |
| SC-HDR15-50 | 3108.11 ± 11.50 [e] | 0.07 ± 0.00 [a] | 0.98 | 376.44 ± 5.95 [d] | 0.11 ± 0.00 [a] | 0.88 | 0.11 [a] |
| SC-HDR15-75 | 3215.11 ± 30.75 [e] | 0.11 ± 0.00 [b] | 0.95 | 473.52 ± 8.06 [e] | 0.16 ± 0.00 [b] | 0.93 | 0.17 [c] |
| SC-HDR20-25 | 68.94 ± 0.47 [a] | 0.17 ± 0.00 [d] | 0.99 | 68.94 ± 0.47 [b] | 0.17 ± 0.00 [b] | 0.99 | 0.34 [e] |
| SC-HDR20-50 | 156.37 ± 1.94 [b] | 0.15 ± 0.00 [c] | 0.96 | 156.37 ± 1.94 [c] | 0.15 ± 0.00 [b] | 0.96 | 0.15 [bc] |
| SC-HDR20-75 | ** | ** | ** | ** | ** | ** | 0.17 [c] |

Results are expressed as the mean ± standard deviation. The different letters within each column are significantly different ($p < 0.05$). Tanδ represents CV < 0.5. ** The samples did not conform to the power-law model.

$k'$ was higher for mayonnaises developed with HDR15, between 2425 and 3215, while for mayonnaises developed with HDR20, lower values (lower than 156) were observed in comparison with control samples (205). In addition, $k''$ was lower than 473, indicating the magnitude of $G'$ and $G''$ at a given frequency. In all cases, $n'$ was less than 0.17, presenting the lowest value for SC-HDR15-25 and SC-HDR15-50, and $n''$ was less than 0.18, presenting the lowest value for SC. For example, $n'$ and $n''$ were closer to zero, meaning no change with frequency [69]. Furthermore, $n''$ was slightly higher than $n'$, denoting a great inner dynamism in the network that tended to flow at a higher frequency, while at

lower frequencies, it tended to present a more solid-like behavior. Samples with low values of $G'$, such as SC and SC-HDR20-25, were shown to have a lower viscoelastic behavior, resulting in a weak structure compared to samples with higher values of $G'$; they were SC-HDR15-25, SC-HDR15-50, and SC-HDR15-75 due to a stronger structure due to the concentration of hydrocolloids that allowed the formation of large aggregates to produce a better viscoelastic behavior.

The loss factor (Tan $\delta$) is used to interpret the viscoelastic behavior at 25 °C, where the purely elastic characteristics are read as $\delta = 0°$ and $G' > G''$, and purely viscous, $\delta = 90°$ and $G'' > G'$ [70]. The loss tangent is a dimensionless measure that expresses the energy lost with respect to the energy stored during a test, which has values from 0 to 1 when the matrix has an elastic character [70]. Figure 8 shows the values of the tangent of the phase angle of mayonnaise with different concentrations of hydrocolloids.

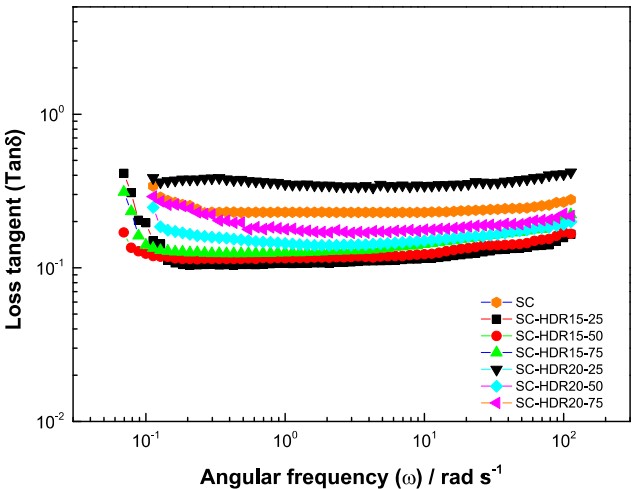

**Figure 8.** The loss tangent of mayonnaise using the hydrocolloids of *Dioscorea rotundata* as a fat substitute.

It can be seen that the values obtained for the elastic modulus ($G'$) were significantly higher than the values obtained for the viscous modulus ($G''$) (Table 4), which means that the results of the tangent of the phase angle were less than 1. This implies that the elastic properties were predominant in the emulsion. According to Figure 7, the SC-HDR20-25 formulation was the most fluid mayonnaise due to the highest test values. The SC, SC-HDR20-50, and SC-HDR20-75 formulas were progressively more fluid, and those with a somewhat solid structure were SC-HDR15-25 and SC-HDR15-50.

According to the results of the viscoelastic behavior of each sample, it is notable that all formulations presented higher values of $G'$, indicating a solid elastic behavior typical of sauces and dressings. Based on this, the use of *Dioscorea rotundata* hydrocolloids is a viable alternative as a substitute for fats at different concentrations in the production of food products. The SC-HDR15-25 sample presented similar physicochemical, rheological, and microstructural properties to SC.

## 4. Conclusions

Hydrocolloids from *Dioscorea rotundata* can be used to reduce the oil content in mayonnaise, preserving the physicochemical properties up to 14 days of storage. The pH did not vary; with respect to color, there was a decrease in luminosity and an increase in parameters a* and b* over time, indicating a decrease in white and an increase in yellow. The microstructure of the emulsions showed droplet sizes less than 6 μm, which contributed to the stability by preventing phase separation in the evaluated period. The rheological study of mayonnaises allowed us to conclude that all samples presented a non-Newtonian flow behavior (shear thinning), which fitted the Sisko rheological model with an average

R$^2 \geq 0.99$. The dynamic viscoelastic properties were characterized by an oscillatory frequency sweep under conditions of low deformation, shown as an increase in the elastic modulus that indicated solid behavior, as the ingredients contributed to maintaining the stability of the samples. The samples showed a plateau zone characteristic of systems such as weak gels adjusted to the power-law model. This work could facilitate the design of reduced-fat mayonnaise as a novel health product to be developed on an industrial scale. According to this, it is concluded that *Dioscorea rotundata* hydrocolloid as a fat substitute at a concentration of HDR15 improved the viscoelastic behavior of mayonnaise.

**Author Contributions:** Conceptualization, L.R.-M., S.E.Q. and L.A.G.-Z.; methodology, L.R.-M. and S.E.Q.; software, L.R.-M., S.E.Q. and L.A.G.-Z.; validation, S.E.Q. and L.A.G.-Z.; formal analysis, L.R.-M., S.E.Q. and L.A.G.-Z.; investigation, L.R.-M., S.E.Q. and L.A.G.-Z.; writing—original draft preparation, L.R-M.; writing—review and editing, L.R.-M. and L.A.G.-Z.; supervision, S.E.Q. and L.A.G.-Z.; project administration, L.A.G.-Z.; funding acquisition, L.A.G.-Z. All authors have read and agreed to the published version of the manuscript.

**Funding:** This research was funded by the Universidad de Cartagena, grant number No. 142-2019.

**Data Availability Statement:** Not Applicable.

**Conflicts of Interest:** The authors declare no conflict of interest.

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
