# Peer review of "Physicochemical, Rheological, and Microstructural Properties of Low-Fat Mayonnaise Manufactured with Hydrocolloids from Dioscorea rotundata as a Fat Substitute"

_processes, doi:10.3390/pr11020492_

Round 1

Reviewer 1 Report

Comment: Line 26-35 this part should be revised. Is the cholesterol content high in mayonnaise?  The sentence should be revised as The high “oil content may cause increase in cholesterol content”.

Comment: what is the novelty of this study ? in introduction section novelty of this study should be clearly explain.

Comment: Table 1 should be revised. All abbreviations should be clearly written at the bottom of table.

Comment: …….two hydrocolloids solutions were prepared with 15%  (HDR15) and 20% (HDR20) to replace the oil in percentages of 25, 50, and 75 %,…….. What is the oil content of the control sample ? This section should be clearly written. The oil content of the other samples should be written not substitute value

Comment: Line 113 What is the criteria for determining the color change ? the basis is control samples ?

Comment. Rheological analysis what is the gap value between the rheometer probe ?

Comment: For the samples HDR15 the L value showed increasing and decreasing trend. For the HDR20 samples, the change in L value showed opposite trend. How can this result be explained? Why the b* value of the samples SC-HDR15-25 is very high ?.

Comment: The graphs of the rheological analyzes are mixed each other. All rheology graphs need to be improved. Also, what is the main reason for the difference between the rheological properties of the samples? Why is there no difference between the SC-HDR15-25 and SC-HDR20 -25 samples, while there is no significant difference between the SC-HDR15-50 sample and the SC-HDR20-50 sample? No significant difference was observed between the SC-HDR15-25 and SC-HDR15-50 samples, how can this be explained? Which factor is main factor affecting rheological parameters oil content or hydrocolloid content? Because different trends was observed.

Comment: The data of the frequency sweep test should be also modeld by power low or other models. The graps of the frequency sweep test should be improved.  The numerical value may be useful than logarithmic scale on the axes. The G’ and G’’ values are indistinguishable and not understand viscoelastic behavior of the samples. The k value of the control samples is higher than other samples with exception of SC-HDR15-75 and SC-HDR20-75. why this trend was not observed in G' values. The G’ value of the samples SC-HDR20-25 and SC-HDR15-25  was 289.1 and 2940 respectively. How can be explained this great difference ?

Comment: According to all these results, which sample can be used instead of the control sample? What is your recommendation?

Author Response

Cartagena de Indias, January 15th, 2023

Manuscript ID: processes-2134457

Title: Physicochemical, rheological and microstructural properties of low-fat mayonnaise using hydrocolloids from Dioscorea rotundata as a fat substitute.

Authors: Leonardo Rojas-Martin, Somaris E. Quintana, Luis A. García-Zapateiro.

Dear editor,

The authors of the article “Physicochemical, rheological, and microstructural properties of low-fat mayonnaise using hydrocolloids from Dioscorea rotundata as a fat substitute”- Manuscript ID: processes-2134457 are grateful for the comments made by the referee and the editor, which allowed us to improve their presentation. Thank you for the kind comments on this paper. The English grammar and style have been revised and hopefully improved. Furthermore, according to the reviewer's suggestion, we agree with the grammar corrections of the reviewer and revised the entire article. The article has been revised with the help of a native English speaker from Cartagena University. Likewise, the new document is sent with changes written in green color.

Answers to the reviewer

Reviewer 1

Comment: Lines 26-35 this part should be revised. Is the cholesterol content high in mayonnaise? The sentence should be revised as the high “oil content may cause increase in cholesterol content”.

Answer: The introduction was improved.

Comment: what is the novelty of this study? in introduction section novelty of this study should be clearly explain.

Answer: The introduction was improved.

Comment: Table 1 should be revised. All abbreviations should be clearly written at the bottom of table.

Answer: The table was improved.

Comment: …….two hydrocolloids solutions were prepared with 15%  (HDR15) and 20% (HDR20) to replace the oil in percentages of 25, 50, and 75 %,…….. What is the oil content of the control sample ? This section should be clearly written. The oil content of the other samples should be written not substitute value

Answer: The method of development mayonnaise was performed.

Comment: Line 113 What is the criteria for determining the color change ? the basis is control samples ?

Answer: The text was improved. The color change was calculated using equation 1 between the control samples and mayonnaises with HDR15 and HDR20.

Comment. Rheological analysis what is the gap value between the rheometer probe ?

Answer: The GAP value was added.

Comment: For the samples HDR15 the L value showed increasing and decreasing trend. For the HDR20 samples, the change in L value showed opposite trend. How can this result be explained? Why the b* value of the samples SC-HDR15-25 is very high ?.

Answer: An explanation of the luminosity of the samples appears in the following text:this is due to the presence of a branched structure of hydrophilic macromolecules that allows the formation of homogeneous emulsions with smaller oil droplets, which generates greater light scattering.

Comment: The graphs of the rheological analyzes are mixed each other. All rheology graphs need to be improved. Also, what is the main reason for the difference between the rheological properties of the samples? Why is there no difference between the SC-HDR15-25 and SC-HDR20 -25 samples, while there is no significant difference between the SC-HDR15-50 sample and the SC-HDR20-50 sample? No significant difference was observed between the SC-HDR15-25 and SC-HDR15-50 samples, how can this be explained? Which factor is main factor affecting rheological parameters oil content or hydrocolloid content? Because different trends was observed.

Answer: All figures were improved. The rheological behavior of the samples is associated with the molecular entaglement, as it explained in L348. The mayonnaises showed a characteristic behavior; for example, at a lower speed, there is a coiling or molecular entanglement in the structure, generating higher viscosity

Comment: The data of the frequency sweep test should be also modeld by power low or other models. The graph of the frequency sweep test should be improved. The numerical value may be useful than logarithmic scale on the axes. The G’ and G’’ values are indistinguishable and not understand viscoelastic behavior of the samples. The k value of the control samples is higher than other samples with exception of SC-HDR15-75 and SC-HDR20-75. why this trend was not observed in G' values. The G’ value of the samples SC-HDR20-25 and SC-HDR15-25 was 289.1 and 2940 respectively. How can be explained this great difference ?

Answer: The power law model was used to adjust the experimental data of G’ and G’’. The figures were improved. Moreover, the analysis was added.

Comment: According to all these results, which sample can be used instead of the control sample? What is your recommendation?

Answer: The analysis of the results was improved. The following text was added: L481. The SC-HDR15-25 sample presented physicochemical, rheological, and microstructural properties similar to those of SC.

Leonardo Rojas-Martin

UNIVERSIDAD DE CARTAGENA

Somaris Elena Quintana Martínez, PhD.

UNIVERSIDAD DE CARTAGENA

Prof. Luis Alberto García Zapateiro, MSc, PhD.

UNIVERSIDAD DE CARTAGENA

Reviewer 2 Report

L12, showed that…

L15, what is the SC1?

L56-57, this sentence is incomplete.

L57-60, this sentence is difficult to be understand. Please rewrite it.

L143, “until” should be replaced by “under”.

L154-158, too long sentence.

L161, 237, what different letters refer to in Figure 1 and 2?

L412-416, please rewrite this sentence.

L417, some important and highlighted conclusions should be presented in the conclusions section, which is different from abstract section.

Author Response

Cartagena de Indias, January 15th, 2023

Manuscript ID: processes-2134457

Title: Physicochemical, rheological and microstructural properties of low-fat mayonnaise using hydrocolloids from Dioscorea rotundata as a fat substitute.

Authors: Leonardo Rojas-Martin, Somaris E. Quintana, Luis A. García-Zapateiro.

Dear editor,

The authors of the article “Physicochemical, rheological, and microstructural properties of low-fat mayonnaise using hydrocolloids from Dioscorea rotundata as a fat substitute”- Manuscript ID: processes-2134457 are grateful for the comments made by the referee and the editor, which allowed us to improve their presentation. Thank you for the kind comments on this paper. The English grammar and style have been revised and hopefully improved. Furthermore, according to the reviewer's suggestion, we agree with the grammar corrections of the reviewer and revised the entire article. The article has been revised with the help of a native English speaker from Cartagena University. Likewise, the new document is sent with changes written in green color.

Answers to the reviewer

Reviewer 2

L12, showed that…

L15, what is the SC1?

L56-57, this sentence is incomplete.

L57-60, this sentence is difficult to be understand. Please rewrite it.

L143, “until” should be replaced by “under”.

L154-158, too long sentence.

L161, 237, what different letters refer to in Figure 1 and 2?

L412-416, please rewrite this sentence.

L417, some important and highlighted conclusions should be presented in the conclusions section, which is different from abstract section.

Answer: The manuscript was improved. The English grammar and style have been revised and hopefully improved, we agree with the grammar corrections of the reviewer and revised the entire article. The article has been revised with the help of a native English speaker from Cartagena University. Redaction was checked. The figures and tables were checked and improved. Conclusions were improved.

Leonardo Rojas-Martin

UNIVERSIDAD DE CARTAGENA

Somaris Elena Quintana Martínez, PhD.

UNIVERSIDAD DE CARTAGENA

Prof. Luis Alberto García Zapateiro, MSc, PhD.

UNIVERSIDAD DE CARTAGENA
